# A Comparison Study of the Western Ontario Rotator Cuff Index, and the Constant–Murley Score with Objective Assessment of External Rotator Muscle Strength and Pain in Patients after Arthroscopic Rotator Cuff Repair

**DOI:** 10.3390/ijerph20136316

**Published:** 2023-07-07

**Authors:** Agnieszka Bejer, Jędrzej Płocki, Mirosław Probachta, Ireneusz Kotela, Andrzej Kotela

**Affiliations:** 1Institute of Health Sciences, College of Medical Sciences, University of Rzeszow, 35-959 Rzeszow, Poland; 2The Holy Family Specialistic Hospital, 36-060 Rzeszow, Poland; 3Department of Physiotherapy, Collegium Medicum, University of Information Technology and Management in Rzeszow, 35-225 Rzeszow, Poland; 4Institute of Health Sciences, Collegium Medicum, The Jan Kochanowski University, 25-317 Kielce, Poland; 5Department of Orthopaedic Surgery and Traumatology, Central Clinical Hospital of the Ministry of Interior, 02-507 Warsaw, Poland; 6Faculty of Medicine, Collegium Medicum, Cardinal Stefan Wyszyński University, 01-938 Warsaw, Poland

**Keywords:** rotator cuff injuries, Western Ontario Rotator Cuff Index, Constant–Murley score, muscle strength, shoulder pain

## Abstract

Although rotator cuff injures are often associated with a limited range of motion and muscle weakness, being able to conduct pain-free and efficient performances of the activities as part of daily living seems to be more important for patients. The aim of this study was to investigate the correlation between two questionnaires—the disease-specific, subjective questionnaire termed the Western Ontario Rotator Cuff Index (WORC), and the shoulder-specific, subjective-objective questionnaire Constant–Murley score (CMS), with the objective assessment of external rotator muscle strength, and the subjective assessment of pain according to the visual analog scale (VAS) in patients after arthroscopic rotator cuff repair. The study was carried out among 47 patients twice—6 and 12 months after surgery, respectively. All patients completed the WORC, the CMS, and the VAS. Isokinetic evaluation of the external rotators was performed using the Biodex 4 ProSystem. The correlations of all assessed muscle strength parameters with both the CMS and the WORC were found to be statistically significant, being mostly average during the 1st examination and mostly strong during the 2nd examination. There was a significant improvement in all assessed tools as a result of the undertaken rehabilitation. There were weak correlations present between changes in the WORC and changes in the external rotator muscle strength, with correlations between WORC-Sport and EXT90^0^-AVERAGE-POWER and PEAK-TORQUE also being found statistically significant. Correlations of changes in the CMS scale with changes in the external rotator muscle strength were weak and statistically insignificant. It seems that the WORC questionnaire can be recommended more for the population after rotator cuff repair, which allows for a reliable assessment of patients’ ability to function and its changes in various areas of life, and at the same time does not require a direct assessment by a clinician or researcher.

## 1. Introduction

The correct functioning of the upper limb depends on a good cooperation of all the structures comprising the shoulder complex. Disturbances in the functioning of a single element in the mechanism of the shoulder complex can result in the impaired manipulation of the entire upper limb [1]. A rotator cuff tear is the most commonly observed and treated tendon injury in adults. The incidence amounts to approximately 30% of the population over the age of 60, and about 62% of the population over the age of 80, respectively [2]. Rotator cuff disorders cover a wide range of injuries, including from tendinopathy to partial tears, as well as complete tears. They are the result of a mechanical injury or a consequence of long-term overloads [3]. Patients with a rotator cuff injury report increasing pain, limitation of the mobility of the shoulder joint, loss in muscle strength of the shoulder girdle, and difficulty with overhead activities, activities of daily living, along with professional and sports activity. They may also report pain when lifting/carrying heavy objects or when they sleep on the side of the lesion [4].

The primary goal of rehabilitation in patients with rotator cuff injures treated both surgically and conservatively is to restore their functionality of the upper limb while simultaneously improving the level of participation in their social lives [5]. Reliable, accurate, and responsive measurement tools are needed to assess the function of the upper limb and to monitor the changes resulting from the intervention undertaken. They allow the clinician to adapt the therapy to the existing deficits and implement modifications depending on the achieved results [6]. Shoulder function tends to be assessed by the means of objective measures, including the range of motion, muscle strength, and endurance [7]. However, the literature indicates that patient-reported outcome measures (PROMs) providing information from the patient’s perspective on their health status are more accurate than the objective measures [8]. Although rotator cuff injures are often associated with a limited range of motion and muscle weakness, pain-free and efficient performance of the activities of daily living seems to be more important for patients [9].

Numerous questionnaires have been developed for measurement of disease-, shoulder-, and upper extremity-specific outcomes. For rotator cuff and other subacromial pathologies, the American Shoulder and Elbow Surgeon (ASES) questionnaire, the Disabilities of the Arm, Shoulder, and Hand (DASH) questionnaire, the Shoulder Pain and Disability Index (SPADI), the Upper Limb Functional Index (ULFI), the Simple Shoulder Test (SST), the Oxford shoulder score (OSS), the Western Ontario Rotator Cuff Index (WORC), and the Constant–Murley score (CMS) are all recommended and frequently used PROMs [6,10,11].

The aim of this study was to investigate the correlation of two questionnaires—the disease-specific, subjective questionnaire Western Ontario Rotator Cuff Index, and the shoulder-specific, subjective-objective questionnaire Constant–Murley score, with the objective assessment of external rotator muscle strength and the subjective assessment of pain according to the visual analog scale (VAS) in patients after arthroscopic rotator cuff repair. Correlations of changes occurring as a result of rehabilitation were assessed using specific questionnaires with changes in muscle strength and pain also being assessed.

The WORC was created and published by Kirkley et al. in 2003, who confirmed its high reliability, validity, and responsiveness among patients with rotator cuff injures. It allows for the subjective and reliable assessment of the symptoms, functioning and quality of life, and verification of the effectiveness of the treatment applied from patients’ perspectives [12]. The CMS was created by Constant and Murley in 1987, and was subsequently modified in 2008 by Constant et al. It was designed to assess shoulder disorders by combining subjective and objective measurements, such as pain, activities of daily living, strength, and the range of motion [13,14]. The use of the CMS is mainly recommended in patients with subacromial pathology, as in this group of patients it shows the best psychometric properties [15].

## 2. Materials and Methods

### 2.1. Study Sample and Study Design

The study was carried out in the period from March 2015 until April 2017, respectively, among the patients operated in the of Holy Family Specialist Hospital in Rudna Mała, Poland due to injury of the tendons of the muscles forming the rotator cuff.

The inclusion criteria included: condition after arthroscopic surgical treatment involving reinsertion of the tendon insertions of the rotator cuff of the shoulder joint, age from 40 to 65 years, respectively, and informed and voluntary consent to participate in the study. The exclusion criteria are: instability of the glenohumeral joint, history of dislocations within the shoulder complex, history of fractures of the proximal end of the humerus, acetabulum, clavicle, or acromion, bilateral rotator cuff damage, occurrence of neurological disorders and deficits and diseases that may affect examination results, and palliative rotator cuff reconstruction.

The examination was performed twice—6 and 12 months after arthroscopic reconstruction of the rotator cuff. During the period between these tests, the patients participated in a rehabilitation program designed by researchers and physiotherapists, based on guidelines from the literature [16,17], and was approved by the orthopedist who had performed the surgery. The protocol for the physiotherapy (0–12 months) was presented to the subjects at an early (in-patient) stage following the surgery. It contained the required guidelines to ensure the balance between the limitations necessary for the healing of the tissues, and activity enabling the gradual restoration of the functions of the shoulder complex. The early stage comprised the passive therapy phase, with a duration of 6–8 weeks, depending on the size of the tear. The next stage involved supported exercise and then active training. Resistance exercises were gradually introduced during the third stage (which began from month 4), provided that the patient regained a non-painful, reasonably complete active range of movement with no compensation. This type of exercise was continued during the next stage, starting from month 6, which was intended to build the patient’s strength and power, as well as endurance of the shoulder complex. Training designed to prepare the patient to safely return to work or active recreational activities was not started until Week 30. Each stage of the protocol was discussed in detail, verified, and adjusted to the specific needs of the patient by one physiotherapist from the specialist hospital in Rudna Mała in the period of weeks 2 and 6, as well as months 3, 6, and 12 after the surgery, respectively. Patients received the outpatient physiotherapy at the place of residence, according to the specified protocol.

### 2.2. Measurements

#### 2.2.1. The Western Ontario Rotator Cuff Index (WORC)

The WORC is a disease-specific and self-reported questionnaire, which was designed for patients with injures of the rotator cuff by Kirkley et al. in 2003. It contains 21 items grouped into five domains—physical symptoms (six items), sport/recreation (four items), work (four items), lifestyle (four items), and emotions (three items). All items have the same weight, and each has a possible score ranging from 0 to 100, respectively, on a 100 mm visual analogue scale (VAS). Therefore, the total scores computed separately for the five domains have the following maximum values: physical symptoms—600; sport and recreation—400; work—400; lifestyle—400; and emotions—300, respectively. The total score in the whole questionnaire is in the range between 0 and 2100, the lowest score corresponding to no symptoms, and the highest score reflecting the worst symptoms possible, respectively. A clinically meaningful result is presented as a percent value corresponding to the numerical result. To calculate the percent value it is necessary to subtract the total score from 2100 (the worst possible result), divide the obtained number by 2100, and then multiply it by 100. The total WORC score ranges between 0% and 100%, the former corresponding to the poorest functional status, and the latter reflecting the highest level of functioning, respectively [12,18].

#### 2.2.2. The Constant–Murley Score (CMS)

The CMS was created by Constant and Murley in 1987 to globally assess shoulder function, and then in 2008 by Constant et al., wherein it was modified and standardized the CMS guidelines. The CMS was designed to evaluate pain and disability using both subjective and objective measures. The maximum total CMS amounting to 100 points (the best condition), consists of 35 points scored in subjective measures and 65 in objective measures, respectively. The subjective patient-based component comprises 5 items assessing pain (with a maximum score of 15 points), and activities of daily living (20 points). The objective clinician-based component (with a maximum score of 65 points) measures the active range of motion—i.e., pain-free forward and lateral elevation, external and internal rotation (40 points), and the isometric force is measured with a dynamometer in 90 degrees of abduction in the shoulder in the plane of the scapula (25 points) [13,14,19].

#### 2.2.3. The Visual Analog Scale (VAS)

The VAS was first used to assess the severity of pain by Hayes and Patterson in 1921. This self-report tool utilizes a 10 cm line representing a continuum of symptoms, ranging from “no pain” (on the left end—at 0 cm) to the “worst pain imaginable” (on the right end of the scale—at 10 cm), respectively. The patient rates his/her pain symptoms by placing a single handwritten mark along the line in a spot reflecting the intensity of the perceived symptoms [20].

#### 2.2.4. Isokinetic Evaluation of the External Rotators Using the Biodex 4 Pro System

##### The Course of the Examination on the Biodex 4 Pro System

The Biodex 4 Pro system was used to measure muscle strength moments under isokinetic conditions. Measurements were carried out in a sitting position in a so-called modified neutral position. The subject was stabilized with two belts crossing over the chest and an additional belt running through the hip girdle in order to eliminate possible compensation from other parts of the body. The examined upper limb (operated) was in the position of 45° abduction in the shoulder joint in the plane of the scapula, i.e., 30° flexion in the horizontal plane, with 90° elbow flexing and the forearm set in a neutral position [21] (Figure 1).

The examination of the external rotators of the shoulder joint was performed under eccentric working conditions. Five repetitions were made at an angular velocity of 90°/s. During the assessment, the subjects were encouraged to develop their maximum speed and strength, which guaranteed the objectivity of the assessment. The test was started from the position of maximum external rotation in a predetermined range of motion. Before starting the study, the subjects performed a standardized warm-up consisting of exercises on a rotor for upper limbs (5 min), active exercises with a gymnastic stick (5 min), and exercises on the test stand, where a trial series consisting of 10 repetitions of an eccentric external rotation at a speed of 180°/s was performed with any patient involvement in the task [22].

##### Evaluated Parameters

The results obtained during the examination allowed the assessment of the strength and the strength-velocity capability of the external rotators of the GH joint, such as [1]:

PEAK TORQUE—the peak value of the maximum torque, expressed in Newton meters (Nm). This is the maximum value of the torque at any time during the test. This value indicates the maximum strength capabilities of the assessed muscle group.

TOTAL WORK—total work performed by the external rotators during the test, expressed in joules (J). This is the value of all the work performed over all iterations of one test. It determines whether and to what extent the assessed individual is able to maintain the peak values of the maximum moments of force for all the repetitions performed.

AVERAGE POWER—average power generated by external rotators, expressed in Watts (W). This is the ratio of the total work performed during the test to the time in which the work was carried out.

### 2.3. Ethics

The study was conducted in accordance with the Declaration of Helsinki and was approved by the Ethics Committee of the University of Rzeszów (protocol code No. 2/03/2015).

### 2.4. Statistical Analyses

The statistical analyses were conducted using R software, version 4.2.2 [23]. Normality of data was assessed with the Shapiro–Wilk test. Spearman’s rank correlations were used to compare the outcome measures. Guilford’s classification was used to assess the strength of the correlation, which was as follows: |r| = 0—no correlation; 0.0 < |r| ≤ 0.1—slight correlation; 0.1 < |r| ≤ 0.3—weak correlation; 0.3 < |r| ≤ 0.5—average correlation; 0.5 < |r| ≤ 0.7—strong correlation; 0.7 < |r| ≤ 0.9—very strong correlation; 0.9 < |r| < 1.0—almost full correlation; and |r| = 1—full correlation. The level of statistical significance was set a priori at *p* ≤ 0.05 [24].

## 3. Results

### 3.1. Participant Characteristics

In total, 89 people after arthroscopic rotator cuff reconstruction were qualified for the study, of which 67 met the inclusion criteria for the study. The second assessment took into account forty-seven patients—ten individuals (14.9%) refused to come for the examination because of the long distance from their place of residence; six individuals (9%) were excluded since they did not participate in the physiotherapy program recommended, despite the deficits affecting the range of motion, muscle strength, and functions of the upper limb; three individuals (4.5%) left Poland, and one person (1.5%) withdrew due to experiencing persistent pain in the shoulder during activities requiring the use of force. The patient demographic and clinical characteristics are presented in Table 1.

The values obtained in the CMS and WORC questionnaires, the VAS scale, and muscle strength measurements from Tests 1 and 2 are presented in Table 2.

### 3.2. Relationships between the CMS and WORC Questionnaires and the Pain (VAS) and Muscle Strength (Biodex)

#### 3.2.1. Test 1

All correlations (except correlations between EXT 90 PEAK TORQUE, and measures of CMS, WORC physical symptoms, WORC sport and WORC emotions) were found to be statistically significant. At the same time, WORC total correlated slightly more strongly than CMS with all the assessed parameters of muscle strength and pain (Table 3).

#### 3.2.2. Test 2

The correlations of all the assessed muscle strength and pain parameters with both the CMS questionnaire and the WORC questionnaire were found to be statistically significant, and at the same time mostly strong. The WORC questionnaire correlated more strongly with the VAS, and the CMS slightly more strongly with the muscle strength. At the same time, there were stronger relationships found between these evaluated parameters than in test 1 (Table 4).

### 3.3. Relationships between Changes as a Result of Rehabilitation in the CMS and WORC Questionnaires and Changes in the Pain (VAS) and Muscle Strength (Biodex)

Between the 1st and 2nd examination, there was a significant improvement observed in all the assessed tools as a result of the undertaken rehabilitation (Table 5).

Changes in the CMS correlated slightly more strongly than changes in the WORC with changes in the VAS, and statistically significantly (except for the VAS and WORC lifestyle). There were weak correlations observed between changes in the WORC and changes in the muscle strength, with correlations between WORC sport and EXT 90 AVERAGE POWER and PEAK TORQUE also being found to be statistically significant. However, correlations of changes in the CMS scale with changes in the muscle strength were weak and statistically insignificant (Table 6).

## 4. Discussion

Rotator cuff injury leads to significant impairments in the activities of daily living, work, and sports, and also deteriorates the quality of life. To assess the condition of the patients, objective measurements were used, such as evaluating the strength of the shoulder muscles, muscle activity using sEMG, and the range of motion. However, over the past few decades, the measurements of upper limb functionality and health-related quality of life has become increasingly important in assessing the effectiveness of orthopedic and physiotherapeutic interventions [25,26,27,28]. For this purpose, many specific patient-reported outcome measures (PROMs) have been developed, meaning therefore the selection of an adequate tool for a clinician or researcher is quite difficult. Angst et al. indicated that there are over 30 different tools by entering the keyword “shoulder” and “assessment” into PubMed [29]. Therefore, the choice of the PROMs used will depend on the purpose for which it will be used (e.g., cross-sectional or longitudinal study) or practical considerations (e.g., time needed for the examination, whether or not there is a need to involve a clinician, the ease of the calculation of the results, costs, and feasibility) [30].

Unger et al. reviewed 81 studies that met the inclusion criteria for the PROMs studies applicable to rotator cuff problems. In this paper, 25 different tools were verified, of which the CMS was the most frequently used one, while the WORC was ranked as seventh [31]. Analyses by Huang H et al. also confirmed the most common use of the CMS in research [32]. Due to the lack of a “gold standard” to assess the functionality and quality of life in patients with rotator cuff injures treated either surgically or conservatively, the authors of this study decided to compare two questionnaires commonly used in this group of patients—the subjective WORC and the subjective-objective CMS, with the objective assessment of shoulder external rotator muscle strength and pain (VAS), which reflected the problems associated with the damaged rotator cuff. The CMS is recommended by the European Society for Surgery of the Shoulder and the Elbow (SECEC-ESSSE) for patients after shoulder surgery. It is an accurate and reliable tool, although the interrater reliability is poor [29,33,34]. Unger et al., indicated that the use of the CMS requires more effort on the part of the clinician than when using the all-subjective PROMs, as muscle strength and the range of motion must be assessed. The inability of patients to perform the assessment themselves and/or remotely is a major disadvantage of the CMS [31]. Measurements using the WORC questionnaire only requires the subjective patient input, and one of its important advantages is its ability to document the impact of a health problem on individual areas of the patient’s life. The results obtained on each subscale may contribute to making prognostic and therapeutic decisions [25]. Compared to the CMS scale, this assessment requires less time, and there is no risk of interrater errors [35]. Additionally, a literature review by Huang et al. confirmed that the WORC questionnaire shows the best psychometric properties according to established criteria among patients with rotator cuff injures [32].

The present study showed that the results obtained in the WORC and CMS questionnaires were mostly significantly correlated with the results obtained during the objective, isokinetic, and eccentric evaluations of external rotator cuff muscle strength assessed using the parameters total work, average power, and peak torque. The strength of correlation in test 1 (6 months after surgery) was mostly average, while it was strong in test 2 (12 months after surgery). Similar results were obtained by MacDermid et al. assessing the relationship between the isokinetic and isometric strengths of the external and internal rotators in patients with rotator cuff pathology (n = 36) with the Shoulder Pain and Disability Index (SPADI). These researchers showed mostly statistically significant relationships, with the strength of correlation from r = −0.25 for the isometric force of the internal rotators to r = −56 for the isometric force of the external rotators, and for the eccentric external rotation the correlations with SPADI were r = −0.46 [26], respectively.

Previous studies have shown that strength is impaired in rotator cuff tendinopathy or trauma and improves with surgery and physical therapy. Measuring the strength of the rotator cuff muscles provides reliable information on the functional integrity of the rotator cuff muscles, which translates into a relationship with the functionality of the upper limb and the quality of life of the patients [26,36].

The key points in the treatment of rotator cuff injuries are pain reduction, improvement in the muscle strength, and function of the upper limb, and thus return to pre-injury levels of everyday activity. The effectiveness of the undertaken interventions, verified by the means of subjective tools evaluating the ability to function in everyday life, ensures that the treatment is focused on the patient and not on the disease [37]. In our study, a significant subjective and objective improvement in the condition of the subjects was obtained in all the assessed measures between the first and second examination. There was a significant relationship observed between the reduction in pain and the increase in shoulder functionality and the quality of life assessed using the CMS and the WORC. On the other hand, no significant correlations were observed between the improvements in the external rotator muscle strength and the improvements in the functionality of the upper limb in the CMS, and associations between the improvements in the external rotator muscle strength with the improvements in the WORC were observed only in the sport domain. Roy et al., and Unger et al., through systematic reviews indicated that the WORC appeared to be one of the most responsive PROMs, and that it is the instrument which can accurately detect change over time, whereas the CMS was determined to be one of the least responsive of the shoulder-specific tools [31,38]. However, the significant changes obtained in our study in individual tools and the lack of correlation observed between the change in the CMS and in most WORC domains and the change in the external rotator muscle strength may indicate that changes in the scope of the undertaken activities or the quality of life also depend on changes in other parameters, and not only on the rotator cuff strength. Kluzek et al. emphasized that PROMs provide important indicators of treatment effectiveness that cannot be captured using objective clinical assessments [39]. Skutek et al. showed that self-administered evaluation instruments like the WORC can be used either alone or in conjunction with more feature-oriented measures like the CMS. Kluzek et al. recommended that PROMs should be used in conjunction with objective measures, which will allow for a more comprehensive assessment of changes in the patient’s condition while conducting research and clinical reporting [35].

The results presented in this study are likely to have been affected by patient selection, treatment, assessment of muscle strength in the shoulder joint limited to one movement only, and a follow-up period. The study group of individuals aged 40–65 years is not representative for the entire population of patients with rotator cuff injury. Furthermore, the study only involved patients who received a surgery. Both the WORC and the CMS questionnaires can be used in other groups of patients with rotator cuff injury. Further research taking into account patients receiving conservative treatments, as well as in individuals below 40 and over 65 years of age, respectively, will make it possible to make recommendations related to PROMs suitable for a wider population of patients with rotator cuff injury or to identify the most accurate tool for the specific group. Another limitation of the current study results from the fact that the comparative analyses of the questionnaires assessing muscle strength only took into account the external rotators. To consider all the possible deficits in the muscle strength of the shoulder joint resulting from the rotator cuff injuries, further research should also investigate the isokinetic strength of internal rotation and forward elevation. This study applied two reliable, recommended, and frequently used PROMs. However, in the future, a greater number of validated questionnaires could be compared with one another in order to verify their applicability across various groups of patients with rotator cuff injuries. Future research can also be designed to compare the condition of the subjects assessed with different questionnaires at different timepoints after the medical interventions. This would make it possible to compare their ability to capture the changes in the patients’ condition.

## 5. Conclusions

Based on the results of this study, it can be indicated that the WORC and CMS questionnaires show a similar assessment of the functionality of the upper limb and its impact on the daily activities in patients after rotator cuff repair. The results obtained in both questionnaires correlated comparably with the objective assessment of external rotator muscle strength, with stronger correlations observed in the period distant from surgery when the muscle strength normalized. Changes in patients’ pain were reflected as significant changes in both questionnaires, while changes in the external rotator muscle strength were only captured by significant changes in the sport domain of the WORC questionnaire. Additionally, changes in the condition of the patients who were identified using the CMS were found to be slightly lower compared to the data reported in the related literature concerning the size of the changes that were deemed as clinically significant for the patient. However, the lack of studies reporting the minimal clinically important difference (MCID) for the WORC in patients with rotator cuff injury receiving surgical treatment with a follow-up shorter than 64 weeks, thereby suggests that it is necessary to carry out further research related to this subject matter.

According to the study, the WORC questionnaire can be recommended more for the patients after rotator cuff reconstruction due to it being more relevant to the external rotator muscle strength. It allows for a reliable assessment of patients’ ability to function and its changes in various areas of life, and at the same time does not require a direct assessment by a clinician nor a researcher.

## Figures and Tables

**Figure 1 ijerph-20-06316-f001:**
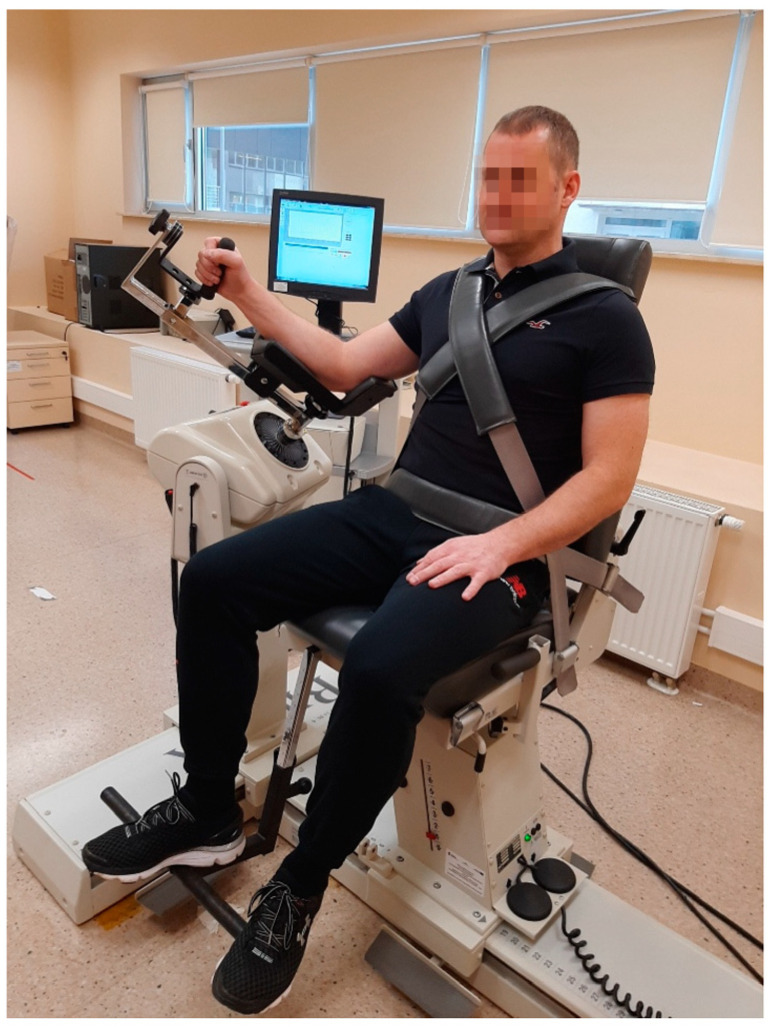
Position taken by the subject during the isokinetic test assessing the external rotators of the shoulder joint (the authors’ own source).

**Table 1 ijerph-20-06316-t001:** Characteristics of the study group.

Parameter		Total (n = 47)
Sex	Woman	15 (31.91%)
	Man	32 (68%)
Age [years]	Mean (SD)	55.87 (5.55)
	Me (Q1–Q3)	57 (52–60)
	Range	40–65
Character of the conditon	Acute	23 (48.94%)
	Chronic	24 (51.06%)
Time from the surgery [months]	Mean (SD)	13.17 (11.1)
	Me (Q1-Q3)	8 (4–24)
	Range	1–48
Operated limb	Right	38 (80.85%)
	Left	9 (19.15%)
Dominating limb	Right	45 (95.74%)
	Left	2 (4.26%)
Degree of injury	Massive rotator cuff injury (≥5 cm in size)	25 (52.08%)
	Supraspinatus muscle injury (full or partial—Ellman grade II or III)	23 (47.91%)

Abbreviations: n, number of subjects; $\bar{x}$, mean; SD, standard deviation; Me, median; Min, minimum value; Max, maximum value; Q1, first quartile; and Q3, third quartile.

**Table 2 ijerph-20-06316-t002:** Data from Test 1 and Test 2 for CMS, WORC, pain (VAS) and muscle strength measurements (Biodex) (n = 47).

Tools	n	$\bar{x}$	SD	Me	Min	Max	Q1	Q3
**Test 1**								
CMS [pts]	47	72.34	12.56	74	42	91	70	79
WORC total [%]	47	72.23	18.21	79	33	99	59	86
WORC physical symptoms [pts]	47	124.06	90.77	109	0	416	62.5	175
WORC sport/recreation [pts]	47	155.3	92.7	154	1	328	84	234
WORC work [pts]	47	148.4	90.57	123	10	317	75	232.5
WORC lifestyle [pts]	47	96.53	83.06	72	0	303	31	149
WORC emotions [pts]	47	58.28	63.37	37	0	207	7.5	77
VAS [0–10]	47	2.5	1.57	3	0	6	2	3
EXT 90 TOTAL WORK [J]	47	69.91	36.39	66.6	26.6	193.4	39	86.6
EXT 90 AVG. POWER [W]	47	12.51	7.91	11.9	1.3	42	6.75	15.3
EXT 90 PEAK TORQUE [Nm]	47	18.61	7.75	17.4	8.5	45.9	12.45	23.4
**Test 2**								
CMS [pts]	47	81.6	12.47	85	45	98	77	89.5
WORC total [%]	47	83.2	16.64	88	33	99.7	73.5	95.5
WORC physical symptoms [pts]	47	78.94	78.63	54	0	336	20.5	106.5
WORC sport/recreation [pts]	47	97.66	92.81	59	0	360	25.5	160
WORC work [pts]	47	89.11	85.68	61	0	320	19	155
WORC lifestyle [pts]	47	56.55	68.48	38	0	277	3.5	85.5
WORC emotions [pts]	47	33.15	51.83	10	0	217	0	30.5
VAS [0–10]	47	1.88	1.67	2	0	6	0	3
EXT 90 TOTAL WORK [J]	47	90.72	43.86	88.4	22.2	191	52.7	120.3
EXT 90 AVG. POWER [W]	47	16.64	9.1	16	2.1	35.7	8.85	24.05
EXT 90 PEAK TORQUE [Nm]	47	20.43	7.62	19.6	9.1	37	13.8	26.85

Abbreviations: n, number of subjects; $\bar{x}$, mean; SD, standard deviation; Me, median; Min, minimum value; Max, maximum value; Q1, first quartile; Q3, third quartile; VAS, visual analog scale; CMS, Constant—Murley score; WORC, Western Ontario Rotator Cuff Index; and EXT, eccentric.

**Table 3 ijerph-20-06316-t003:** Correlations between CMS, WORC, pain (VAS), and muscle strength (Biodex) in Test 1.

	CMS	WORC Total	WORC Physical Symptoms	WORC Sport/Recreation	WORC Work	WORC Life Style	WORC Emotions
**VAS**	r = −0.508, *p* < 0.001 *	r = −0.567, *p* < 0.001 *	r = 0.504, *p* < 0.001 *	r = 0.557, *p* < 0.001 *	r = 0.535, *p* < 0.001 *	r = 0.452, *p* = 0.001 *	r = 0.465, *p* = 0.001 *
**EXT 90 TOTAL WORK**	r = 0.432, *p* = 0.002 *	r = 0.531, *p* < 0.001 *	r = −0.487, *p* = 0.001 *	r = −0.438, *p* = 0.002 *	r = −0.533, *p* < 0.001 *	r = −0.489, *p* < 0.001 *	r = −0.431, *p* = 0.002 *
**EXT 90 AVERAGE POWER**	r = 0.394, *p* = 0.006 *	r = 0.485, *p* = 0.001 *	r = −0.427, *p* = 0.003 *	r = −0.393, *p* = 0.006 *	r = −0.479, *p* = 0.001 *	r = −0.47, *p* = 0.001 *	r = −0.422, *p* = 0.003 *
**EXT 90 PEAK TORQUE**	r = 0.203, *p* = 0.171	r = 0.349, *p* = 0.016 *	r = −0.268, *p* = 0.069	r = −0.229, *p* = 0.122	r = −0.413, *p* = 0.004 *	r = −0.36, *p* = 0.013 *	r = −0.266, *p* = 0.07

Abbreviations: VAS, visual analog scale; CMS, Constant—Murley score; WORC, Western Ontario Rotator Cuff Index; EXT, eccentric; and r, Spearman’s correlation coefficient; * statistically significant relationship (*p* < 0.05).

**Table 4 ijerph-20-06316-t004:** Correlations between CMS, WORC, pain (VAS), and muscle strength (Biodex) in Test 2.

	CMS	WORC Total	WORC Physical Symptoms	WORC Sport/Recreation	WORC Work	WORC Life Style	WORC Emotions
**VAS**	r = −0.565, *p* < 0.001 *	r = −0.771, *p* < 0.001 *	r = 0.803, *p* < 0.001 *	r = 0.672, *p* < 0.001 *	r = 0.72, *p* < 0.001 *	r = 0.743, *p* < 0.001 *	r = 0.605, *p* < 0.001 *
**EXT 90 TOTAL WORK**	r = 0.616, *p* < 0.001 *	r = 0.596, *p* < 0.001 *	r = −0.502, *p* < 0.001 *	r = −0.515, *p* < 0.001 *	r = −0.6, *p* < 0.001 *	r = −0.555, *p* < 0.001 *	r = −0.486, *p* = 0.001 *
**EXT 90 AVERAGE POWER**	r = 0.626, *p* < 0.001 *	r = 0.611, *p* < 0.001 *	r = −0.512, *p* < 0.001 *	r = −0.536, *p* < 0.001 *	r = −0.6, *p* < 0.001 *	r = −0.582, *p* < 0.001 *	r = −0.523, *p* < 0.001 *
**EXT 90 PEAK TORQUE**	r = 0.538, *p* < 0.001 *	r = 0.523, *p* < 0.001 *	r = −0.406, *p* = 0.005 *	r = −0.444, *p* = 0.002 *	r = −0.525, *p* < 0.001 *	r = −0.488, *p* < 0.001 *	r = −0.474, *p* = 0.001 *

Abbreviations: VAS, visual analog scale; CMS, Constant—Murley score; WORC, Western Ontario Rotator Cuff Index; EXT, eccentric; and r, Spearman’s correlation coefficient; * statistically significant relationship (*p* < 0.05).

**Table 5 ijerph-20-06316-t005:** Significance of changes in the individual research tools between the successive studies.

Tools 12 Months vs. 6 Months after Reconstruction	$\bar{x}$	*p*
CMS [pts]	9.26	0.0000 *
WORC Total [%]	10.97	0.0000 *
WORC Physical symptoms [pts]	−45.12	0.0000 *
WORC Sport/recreation [pts]	−57.64	0.0000 *
WORC Work [pts]	−59.29	0.0000 *
WORC Life style [pts]	−39.98	0.0000 *
WORC Emotions [pts]	−25.13	0.0004 *
VAS [0–10]	−0.62	0.0188 *
EXT 90 TOTAL WORK [J]	20.81	0.0000 *
EXT 90 AVG. POWER [W]	4.13	0.0000 *
EXT 90 PEAK TORQUE (Nm)	1.82	0.0034 *

Abbreviations: VAS, visual analog scale; CMS, Constant—Murley score; WORC, Western Ontario Rotator Cuff Index; and EXT, eccentric; $\bar{x}$, mean; * statistically significant relationship (*p* < 0.05).

**Table 6 ijerph-20-06316-t006:** Correlations between changes in the CMS and WORC, and changes in pain (VAS) and muscle strength (Biodex) between Tests 1 and 2.

	CMS	WORC Total	WORC Physical Symptoms	WORC Sport	WORC Work	WORC Life style	WORC Emotions
**VAS**	r = −0.447, *p* = 0.002 *	r = −0.43, *p* = 0.003 *	r = 0.389, *p* = 0.033 *	r = 0.43, *p* = 0.003 *	r = 0.446, *p* = 0.002 *	r = 0.251, *p* = 0.088	r = 0.311, *p* = 0.033 *
**EXT 90 TOTAL WORK**	r = −0.028, *p* = 0.853	r = −0.155, *p* = 0.299	r = 0.123, *p* = 0.409	r = 0.269, *p* = 0.068	r = 0.129, *p* = 0.388	r = 0.013, *p* = 0.928	r = 0.162, *p* = 0.278
**EXT 90 AVERAGE POWER**	r = −0.044, *p* = 0.77	r = −0.208, *p* = 0.161	r = 0.244, *p* = 0.099	r = 0.301, *p* = 0.04 *	r = 0.186, *p* = 0.211	r = 0.039, *p* = 0.793	r = 0.172, *p* = 0.247
**EXT 90 PEAK TORQUE**	r = −0.054, *p* = 0.717	r = −0.231, *p* = 0.118	r = 0.254, *p* = 0.085	r = 0.365, *p* = 0.012 *	r = 0.162, *p* = 0.277	r = 0.076, *p* = 0.611	r = 0.217, *p* = 0.144

Abbreviations: VAS, visual analog scale; CMS, Constant—Murley score; WORC, Western Ontario Rotator Cuff Index; EXT, eccentric; and r, Spearman’s correlation coefficient; * statistically significant relationship (*p* < 0.05).

## Data Availability

The data that support the findings of this study are available from the corresponding author upon reasonable request.

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
