# Peer review of "A Comparison Study of the Western Ontario Rotator Cuff Index, and the Constant–Murley Score with Objective Assessment of External Rotator Muscle Strength and Pain in Patients after Arthroscopic Rotator Cuff Repair"

_ijerph, 2023, doi:10.3390/ijerph20136316_

Round 1

Reviewer 1 Report

The article title: "A comparison study of the Western Ontario Rotator Cuff Index, and the Constant-Murley Score with objective assessment of rotator cuff muscle strength and pain in patients after arthroscopic reconstruction"

The aim of the study was to investigate the correlation of two questionnaires - the disease-specific, subjective questionnaire Western Ontario Rotator Cuff Index (WORC), and the shoulder-specific, subjective-objective questionnaire Constant-Murley Score (CMS) with objective assessment of rotator cuff muscle strength and subjective assessment of pain according to the Visual Analog Scale (VAS) in patients after arthroscopic reconstruction.

It seems that the WORC questionnaire can be more recommended for the population after rotator cuff reconstruction, which allows for a reliable assessment of patients' ability to function and its changes in various areas of life, and at the same time does not require a direct assessment by a clinician or researcher. In the conclusion stays that the WORC questionnaire can be more recom-mended for the population after rotator cuff reconstruction, which allows for a reliable assessment of patients' ability to function and its changes in various areas of life, and at the same time does not require a direct assessment by a clinician or researcher.

Reviewer 2 Report

Overall some proofreading for clarity and adding a few images of the methods would be helpfu.

Intro: The motivation to compare the two outcome measures should be presented earlier in the paper to set up the background of the purpose of the study.

Methods: I'm confused how the external rotators were tested eccentrically while starting in maximum IR. A picture of the setup for the biodex would be helpful as well.

A greater explanation of the course of care would be helpful as well as access to the rehab protocol. Did these subjects get guided rehabilitation by a physio?

Results: 20/67 participants resigning is a near 30% loss. Is there a reason for this?

What are the criteria for the distinction between massive RC injury and Supraspinatus muscle injury? Were all rotator cuff muscle injuries included and treated the same? Is there a reason why there is a relatively small age range for the inclusion? This seems to indicate that the injuries are less trauma induced and more degenerative in nature. This is somewhat covered in the acute/chronic demographic but this isn't clearly defined. 

There is limited discussion of the limitations or future directions of this research including only looking at one other outcome measure and limiting it to post op patients within a 25y age window

Conclusion: Is the difference noted in the WORC clinically significant or meaningful to the patient? The differences seem small but the benefit seems like the WORC is easier to administer with less risk of error.

Reviewer 3 Report

Assessment of functional improvement of shoulder and adaptability of daily activates after rotator cuff surgery is essential for clinicians. Here authors compare two most utilized tools, CMS and WORC score and try to find the correlation between rotator cuff muscle strength and pain score. The conclusion about weak correlation between subjective score and recovery of muscle strength has been elucidated. However, there are a vital question about the results of the research.

That is authors use Biodex 4 Pro System to evaluate the muscle strength, they put the examined upper limb in the position of 45° abduction and test the external rotation work and power torque. The predominant muscles responsible for external rotation is infraspinatus and teres minor muscle. So, in this situation, the Biodex revealed the result of strength only of infraspinatus and teres minor muscle, not the whole rotator cuff muscle. My suggestion is that author should revised the title to Objective Assessment of External Rotator Cuff Muscle Strength not the whole rotator cuff muscle. Authors only detected the external rotation force, there are still internal rotation and forward elevation forces to be tested if they want to make a conclusion like their original research. The article authors cited from MacDermid et al., they assessed relationship between the isokinetic and isometric strength of the external and internal rotators in patients with rotator cuff pathology. Such conclusion is more solid than the present research.

Round 2

Reviewer 3 Report

According to the study, WORC questionnaire can be more recommended for the patients after rotator cuff reconstruction due to being more relavent to external rotator muscle strength. That can be useful informtion for clinician to adapt this patient reported outcom (PROM) survey.

Author Response

Dear Reviewer,

We would like to express our thanks for reviewing our article titled ‘A Comparison Study of the Western Ontario Rotator Cuff Index, and the Constant-Murley Score with Objective Assessment of External Rotator Muscle Strength and Pain in Patients after Arthroscopic Rotator Cuff Repair’, and for the invaluable remark: „According to the study, WORC questionnaire can be more recommended for the patients after rotator cuff reconstruction due to being more relavent to external rotator muscle strength. That can be useful informtion for clinician to adapt this patient reported outcom (PROM) survey”.

We agree that this might be useful informtion for clinicians. So that, the conclusions have been revised as per your suggestion:

Lines 408-410: According to the study, the WORC questionnaire can be more recommended for the patients after rotator cuff reconstruction due to being more relevant to external rotator muscle strength.

With kind regards,

Authors